# Disproportionate CH_4_ Sink Strength from an Endemic, Sub-Alpine Australian Soil Microbial Community

**DOI:** 10.3390/microorganisms9030606

**Published:** 2021-03-15

**Authors:** Marshall D. McDaniel, Marcela Hernández, Marc G. Dumont, Lachlan J. Ingram, Mark A. Adams

**Affiliations:** 1Centre for Carbon Water and Food, Sydney Institute of Agriculture, University of Sydney, Brownlow Hill 2570, Australia; lachlan.ingram@sydney.edu.au (L.J.I.); maadams@swin.edu.au (M.A.A.); 2Department of Agronomy, Iowa State University, Ames, IA 50011, USA; 3Department of Biogeochemistry, Max Planck Institute for Terrestrial Microbiology, D-35037 Marburg, Germany; M.G.Dumont@soton.ac.uk; 4School of Environmental Sciences, Norwich Research Park, University of East Anglia, Norwich NR4 7TJ, UK; 5School of Biological Sciences, University of Southampton, Southampton SO17 1BJ, UK; 6School of Science, Engineering and Technology, University of Swinburne, Melbourne 3122, Australia

**Keywords:** 16S rRNA, carbon dioxide, methane, methanotroph, methanogen, Methylomirabilis, USCα, USC-alpha, *pmoA*

## Abstract

Soil-to-atmosphere methane (CH_4_) fluxes are dependent on opposing microbial processes of production and consumption. Here we use a soil–vegetation gradient in an Australian sub-alpine ecosystem to examine links between composition of soil microbial communities, and the fluxes of greenhouse gases they regulate. For each soil/vegetation type (forest, grassland, and bog), we measured carbon dioxide (CO_2_) and CH_4_ fluxes and their production/consumption at 5 cm intervals to a depth of 30 cm. All soils were sources of CO_2_, ranging from 49 to 93 mg CO_2_ m^−2^ h^−1^. Forest soils were strong net sinks for CH_4_, at rates of up to −413 µg CH_4_ m^−2^ h^−1^. Grassland soils varied, with some soils acting as sources and some as sinks, but overall averaged −97 µg CH_4_ m^−2^ h^−1^. Bog soils were net sources of CH_4_ (+340 µg CH_4_ m^−2^ h^−1^). Methanotrophs were dominated by USCα in forest and grassland soils, and *Candidatus* Methylomirabilis in the bog soils. *Methylocystis* were also detected at relatively low abundance in all soils. Our study suggests that there is a disproportionately large contribution of these ecosystems to the global soil CH_4_ sink, which highlights our dependence on soil ecosystem services in remote locations driven by unique populations of soil microbes. It is paramount to explore and understand these remote, hard-to-reach ecosystems to better understand biogeochemical cycles that underpin global sustainability.

## 1. Introduction

Counteracting biogeochemical processes that consume or produce greenhouse gases (GHGs) regulates whether soils act as net sources or sinks. The magnitude and spatial distributions of these competing processes—whether across landscapes, with soil depth [1,2,3], or even within soil aggregates [4,5,6]—determine net release or uptake of GHGs. Soils, especially upland and older soils, are nearly always sources of carbon dioxide (CO_2_) to the atmosphere due to high production of CO_2_ via heterotrophic decomposition and root respiration, which overwhelms slow rates of autotrophic CO_2_ consumption [7]. On the other hand, soils can routinely be either sources or sinks for methane (CH_4_). In some cases, soils can switch from being a net source to a net sink for CH_4_ in a matter of minutes to hours [8,9,10]. Similarly, lateral distances of less than a meter may be enough for a soil to switch from a sink to a source of GHGs or vice versa [11,12,13].

Methane is a potent greenhouse gas, 34 times as potent as CO_2_, and is responsible for ~17% of anthropogenic warming [14]. Soil sources and sinks for CH_4_ are controlled by abundance and composition of specific microbial communities [15,16], but also regulated by abiotic factors [17,18,19]. Oxidation of CH_4_ to microbial biomass and CO_2_ is, for example, restricted to distinct groups of specialized methanotrophic microorganisms. Aerobic methanotrophs belonging to the α-proteobacteria and γ-proteobacteria groups have been studied for decades and detected across a wide range of habitats [20]. Groundbreaking research in recent years has uncovered previously uncharacterized methanotrophs, including acidophilic Verrucomicrobia from the family Methylacidiphilaceae [21] and intraoxygenic methanotrophs, *Candidatus* Methylomirabilis oxyfera, from the NC10 phylum [22]. In addition, more recent studies have identified USCα methanotrophs, which are specialized atmospheric CH_4_-consuming bacteria whose activity has been known for decades but had evaded efforts to be isolated or characterized until recently [23,24,25]. The first step in biochemical pathways of CH_4_ oxidation is catalyzed by methane monooxygenase (MMO) enzymes. The *pmoA* gene encoding a subunit of the membrane-bound MMO (pMMO) enzyme is present in most aerobic methanotrophs and is frequently used as a genetic marker.

Methanogenesis in soil typically requires anaerobic and low redox conditions, with depletion of strong electronegative terminal electron acceptors, e.g., nitrate and iron. Methanogens belong to the Euryarchaeota phylum, and produce CH_4_ either from CO_2_ and H_2_ (hydrogenotrophic methanogenesis), acetate (acetoclastic methanogenesis), formate, or methylated one-carbon (C_1_) compounds [26]. All known methanogenic pathways include the methyl-coenzyme M reductase enzyme encoded by the *mcrA* gene, which is used as a genetic marker for methanogens in the environment. A few recent studies have shown that even water-logged soil profiles tend to have an aerobic upper layer that, via its capacity for CH_4_ oxidation, helps mitigate CH_4_ release to the atmosphere from lower soil depths. In some cases, this layer can remove 80–97% of CH_4_ produced at lower depths [2,27,28,29]. In the field, and at ecosystem spatial scales, our understanding of the drivers of soil GHG fluxes is profoundly limited due in part because of large spatiotemporal variability and unclear role of key members of the microbial community.

Local soil and vegetation gradients provide an opportunity to explore mechanisms that regulate soil GHG fluxes [30,31,32]. Soil biogeochemical conditions (e.g., microbial substrates, reduction-oxidation potential, and soil pH) can vary dramatically over relatively short distances despite more-or-less constant climate. We used a forest-grassland-bog gradient located within a sub-alpine region of southeastern New South Wales (Figure 1), Australia to examine the drivers of variation in soil microbial communities, and CO_2_ and CH_4_ fluxes. These soils are of particular interest given their previously observed, rapid rates of CH_4_ oxidation in forest and grassland soils (unpublished data) and likely biogeochemical sensitivity to climate change. This endemic, mosaic ecosystem is particularly prone to wildfires in summer, while being snow covered in winter [33,34]. Climate change may increase the frequency and intensity of wildfires and decrease snow cover, having unknown effects on the CH_4_ sink strength of these soils. Our objective was to determine the role of soil microbial communities in regulating the production/consumption of CH_4_ across this unique soil–vegetation gradient.

## 2. Materials and Methods

### 2.1. Study Site Characteristics

Our study sites were located in an area known locally as the Snowy Plains, within the Snowy Mountains region of southern New South Wales. The Snowy Plains form part of the Australian Alps montane grasslands and lie adjacent to the Kosciuszko National Park (36°10′ S, 148°54′ E; Figure 1). The elevation range of the sampling area within the Snowy Plains was 1471 to 1677 m above sea level. The mean annual temperature and precipitation are 6.4 °C and ~1600 mm, respectively [33]. The plains are typically snow covered for 2–3 months of the year.

The area is a mosaic ecosystem containing a mixture of bog, grassland, and forest and has been well described elsewhere [34]. Alpine humus soils (Chernic tenosol) in the region are ~400 million years old and derived from glacial moraines of Silurian Mowomba granodiorite [35]. They show little horizon development in the top 30 cm (Appendix A). These soils are mostly sandy loam in texture, and have pH ranges from ~5.3–5.7 down to 30 cm, across all sites (Appendix A). Bogs in this region are dominated by *Sphagnum* spp. and some sparse grass cover (*Poa* spp.). Grasslands are dominated mostly by *Poa hiemata* and *Poa costiniana*. Forests are dominated by Snow Gums (*Eucalyptus pauciflora*), with the N-fixing shrub *Bossiaea foliosa*, and some *Poa* spp. as understory and ground layers.

### 2.2. Experimental Design, Soil Sampling, and Gas Sampling

We used four topographic transects (from bogs and grasslands in the lowest part of the landscape, to forests in upland areas) to guide our soil and GHG (i.e., CO_2_ and CH_4_) sampling (Figure 1). After each sampling location was determined, a 15 cm diameter (5 cm deep) PVC collar was installed for soil GHG sampling a minimum of 1 month prior to sampling, in order to preclude artifacts introduced by recent disturbance. We measured greenhouse gas fluxes in situ and collected soils on 17 February, 25 May, 22 September, and 23 November in 2015. Four replicates for each soil/vegetation type were measured at each time point. In-depth microbial sampling and analyses, however, were conducted only on soils collected on 17 February; the time of year when soils were warmest and when CH_4_ production and uptake peaked (based on prior gas measurements).

All four sampling dates consisted of paired in situ GHG flux measurements combined with immediate soil sampling for laboratory analyses at each of the 12 sites (Figure 1). We first placed a 3.2 L, vented, PVC chamber over the collar. Four gas samples were collected every 10–15 min for 1 h and directly injected into a Labco Exetainer vial. Concurrent measurements of soil moisture (Theta Probe, Delta-T DevicesTheta Probe, Cambridge, UK) and temperature (Novel Ways, Hamilton, New Zealand) were taken in triplicate and averaged for each measurement location (7 cm depth).

Immediately after completing in situ gas sampling, the chambers were removed, and a 5-cm-diameter, 30-cm-deep soil core was taken directly in the center of the collar. These cores were used for determining ancillary soil properties, microbial community analyses, and to measure net CO_2_ production and CH_4_ production/consumption in the laboratory. Soils were processed as quickly as possible and consistently across all soils. We sought to limit artifacts that might result from soil exposure to O_2_. The 30 cm soil cores were stratified and dissected in the field with a clean knife at 5 cm intervals (0–5, 5–10, 10–15, 15–20, 20–25, and 25–30). At each depth, a subsample (~3–5 g of soil) was immediately placed into a 2 mL Eppendorf tube, then placed in liquid nitrogen and stored at −80 °C until DNA was extracted. The remainder of the soil core was placed in a protective PVC sleeve, and then stored in an iced cooler until reaching the laboratory.

The 5-cm subsections were transferred to 1 L jars, where they were briefly incubated at near-ambient air temperature at the time of collection in order to further characterize CO_2_ and CH_4_ production and consumption occurring at each depth (*sensu* [1]; Appendix A). The 5-cm subsections were placed into separate jars, and the jars were flushed with ambient air and sealed with lids with Luer-lock access ports. Four gas samples were taken every 9 h during the 3-day incubations. Once extracted, headspace gas samples were added directly to 5.9 mL Labco Exetainer vials. Once the incubations were completed, the soils were air dried for physical and chemical analyses. The GHG concentrations were analyzed on an Agilent 7890 gas chromatograph equipped with FID, EID, and a Gerstel MPS-2XL autosampler. We used a ShinCarbon ST column (Restek) to separate gases of interest.

### 2.3. Soil Processing and Analyses

After incubation, soils from the six 5-cm subsections were sieved (2 mm) and rocks and roots were removed and weighed. A subsample of each soil was ground for total C and N analysis on a TruSpec Elemental Analyzer (LECO, St. Joseph, MI, USA). Soil texture was analyzed by using the hydrometer method [36]. Soil inorganic N (ammonium and nitrate) was measured by extracting 5 g of soil with 40 mL of 0.5 M K_2_SO_4_ shaken for 1 h and filtered through Whatman #1 paper. These extracts were analyzed on a Lachat Injection-flow Analyzer according to standard methods (Lachat Instruments, Loveland, CO, USA). Dissolved organic C and N were analyzed on the same extracts on a Shimadzu TOC-N Analyzer (Shimadzu Scientific Instruments Inc., Columbia, MD, USA). Electrical conductivity and pH were measured on SevenMulti probe (MettlerToledo, Columbus, OH, USA) with a 1:1 (w:w) ratio with de-ionized water. Heavy elements were analyzed by X-ray fluorescence using a Niton XL3t Ultra Analyzer (Thermo Scientific, Waltham, MA, USA).

### 2.4. DNA Extraction and Quantitative PCR

DNA was extracted using the NucleoSpin^®^ Soil kit (Macherey-Nagel, Düren, Germany), which disrupts microbial cells by bead beating (30 s at 5.5 m s^−1^). DNA purity and quantity were determined using a NanoDrop^®^ Spectrophotometer ND-1000 (Thermo Fisher Scientific, Waltham, MA, USA). DNA at a concentration of 10 ng μL^−1^ was stored at −20 °C for further molecular analysis. The abundances of bacterial and archaeal 16S rRNA genes as well as *pmoA* genes were quantified using an iCycler Instrument (BioRad, Hercules, CA, USA). For all assays, standards containing known numbers of DNA copies of the target gene were used. qPCR conditions for archaeal and bacterial 16S rRNA genes were based on dual-labeled probes. For bacterial 16S rRNA genes, primers Bac338F and Bac805R, and probe Bac516P were used [37]. For archaeal 16S rRNA genes, primers Arc787F and Arc1059R, and probe Arc915P were used [37]. Conditions for both genes were as follows: 0.5 μM of each primer, 0.2 μM of the dual-labeled probe, 3 µL of template (diluted 10-fold from original DNA), 4 mM MgCl_2_ (Sigma-Aldrich, St. Louis, MO, USA) and 12.5 µL of JumpStart Ready Mix (Sigma-Aldrich) in a final volume of 25 µL. The volume of 1 µL of UltraPure non-acetylated BSA (0.8 µg/µL, Thermo Fisher Scientific) was added to archaeal 16S rRNA gene reactions. The program used for both assays included an initial denaturation of 94 °C for 5 min, followed by 35 cycles of 95 °C for 30 s and 62 °C for 60 s extension and signal reading [37]. The qPCR for *pmoA* genes used a SYBR Green method with primers A189F/mb661R [38,39]. Each qPCR tube contained 0.667 µM of each primer, 3 µL of template, 4 mM MgCl_2_ (Sigma-Aldrich), 0.25 µL of FITC (1:1000, Sigma-Aldrich), 12.5 µL of SYBR Green JumpStart Taq Ready Mix (Sigma-Aldrich) and 0.6 µL of UltraPure non-acetylated BSA (0.5 µg/µL, Thermo Fisher Scientific) in a final volume of 25 µL. The qPCR program included an initial denaturation of 94 °C for 6 min, followed by 45 cycles of 94 °C for 25 s, 65.5 °C for 20 s, 72 °C for 35 s, plus 72 °C for 10 s for the plate read. A final melting curve was included as follows: 100 cycles of 75–94.8 °C for 6 s, +0.2 °C cycle^−1^ [39]. Purified PCR products from pure cultures were used as standards and no template negative control reactions were included by adding water instead of template. qPCR efficiencies of 99.6% for bacterial 16S rRNA genes, 78.8–84.9% for archaeal 16S rRNA genes, and 77.2–78.5% for *pmoA* genes were obtained, all with R^2^ values > 0.99. Technical duplicates were performed for each of the samples.

### 2.5. Illumina Amplicon Library Preparation and Sequencing

MiSeq Illumina sequencing was performed for total 16S rRNA and *pmoA* genes. PCR primers 515F and 806R targeting the V4 region of the 16S rRNA gene (approximately 250 bp) were used [40] with an initial denaturation at 94 °C for 5 min, followed by 28 cycles of 94 °C for 30 s, 50 °C for 30 s, and 68 °C for 30 s and a final elongation at 68 °C for 10 min [41]. The amplification of *pmoA* genes was performed via a semi-nested PCR approach using the primers A189F/A682R for the first round PCR [42] as follows: 94 °C for 3 min followed by 30 cycles of 94 °C for 45 s, 62 to 52 °C (touchdown 1 °C per cycle) for 60 s, 68 °C for 3 min, and a final elongation of 68 °C for 10 min [38]. Aliquots of the first round of PCR (0.5 μL) were used as the template in the second round of PCR using the primers A189f, A650r and mb661r in a multiplex PCR as follows: 94 °C for 3 min followed by 25 cycles of 94 °C for 45 s, 56 °C for 60 s and 68 °C for 1 min, and a final elongation of 68 °C for 10 min [38]. Individual PCRs contained a 6 bp molecular barcode integrated into the forward primer. Amplicons were purified using a PCR cleanup kit (Sigma) and quantified using a Qubit 2.0 fluorometer (Invitrogen). An equimolar concentration of the samples was pooled for each of the genes and sequenced on separate runs using 2 × 300 bp MiSeq paired-end protocol. Library preparation and sequencing was performed at the Max Planck Genome Centre (MPGC), Cologne, Germany. Appendix A summarizes primer sequences for both genes and barcode sequences for each of the samples.

### 2.6. Bioinformatics, Data Processing, GIS Modeling, and Statistical Analyses

For 16S rRNA genes, quality filtering and trimming of forward and reverse adaptors from the sequences were performed with cutadapt [43]. Forward and reverse reads were merged using the usearch fastq_mergepairs command [44]. For the *pmoA* gene, one-end run was performed, and the forward adaptor was trimmed using cutadapt. Downstream processing was performed with UPARSE [44] and UCHIME pipelines [45] following the steps detailed in Reim et al. [46]. For 16S rRNA genes, a representative sequence of each operational taxonomic unit (OTU) was classified based on the SILVA-132 16S rRNA gene database using the naïve Bayesian classifier (bootstrap confidence threshold of 80%) in mothur [47]. The *pmoA* genes were classified using the same method, but using the *pmoA* database [48]. Sequence data were deposited in the NCBI Sequence Read Archive (SRA) under accession number PRJNA384296.

A USCα 16S rRNA gene sequence has recently been identified [23], which enabled us to search these sequences in our dataset. The 16S rRNA genes were identified by standalone BLAST against the 16S rRNA OTUs using the USCa_MF sequence (Genbank ID MG203879). Those OTUs with percent ID > 98% relative to the USCa_MF were positively identified as USCα. The relative abundance of USCα in each of the samples could then be calculated from the OTU table.

Greenhouse gas fluxes (both field and incubation) were calculated using linear regression, or as the change in GHG concentrations over the time. Data were screened for normality and heterogeneity of variances, and when not conforming, log transformed [49] for statistical analyses (all CO_2_ and gene abundance data). All univariate statistics were conducted with R software version 3.0.1 [50]. We used linear and non-linear correlation amongst variables to assess whether there were relationships between greenhouse gas production and microbial community, and selected the best-fit model according to highest r^2^ value and lowest *p*-value. Comparisons of variance among soil/vegetation types (bog, grassland, forest) were completed using 1-way ANOVA (α = 0.05) with transect considered random, and depth not included in interactions since depth effects are not independent. Post hoc tests were completed using *emmeans*, adjusted using Tukey’s test for multiple comparisons. We used the *vegan* package [51] for multivariate statistics to analyze the 16S rRNA Illumina data. Non-metric multidimensional scaling (NMDS) was performed using the *decostand* function for ordination of Hellinger distances. Influence of environmental variables on the total diversity of 16S rRNA and *pmoA* genes was analyzed by the *envfit* function (vegan package in R, permutations = 999). Heatmaps were constructed with the *gplots* package [52]. Principal component analysis (PCA) of the Hellinger-transformed data was performed using the *prcomp* function. The OTUs explaining most of the differences between samples were defined as the ten OTUs contributing the largest absolute loadings in the first and second dimensions of the PCA, obtained from the rotation output file [7]. For hierarchical clustering of the distance matrix, we used the “ward.D2” method and *hclust* function. The heatmap was constructed using the *heatmap.2* function in the *ggplot* package [7].

We developed a model to estimate annual CH_4_ production across the Australian Alps based on a number of geographic information system (GIS) datasets. The datasets included elevation and aspect (30 m pixel) as well as daily estimates of maximum and minimum air temperature [53,54] and soil moisture (0–0.1 m; Frost et al., 2018). The climate data rasters were all based on a 5 km pixel. For each of the days for that CH_4_ fluxes were determined in the field, the location of 12 sites was used to extract data from each of the raster datasets using ArcGIS (V10.8, ESRI Systems, Redlands, CA, USA). These data were then used to develop a linear regression model using R (v 4.0.2). The final linear regression model (*p* < 0.0001, F-statistic = 14.95, adjusted r^2^ = 0.77) included the main factors (forest, grassland, and bog) as well as soil moisture and maximum air temperature and their interactions. The spatial extent of each ecosystem across the Australian Alps bioregion [55] was based on tree cover [56] and mapped hydrological flows. Forest area was identified with ≥20% projected foliage cover, grassland area with <20% projected foliage cover, and bog identified as areas within 1 m of a hydrological flow surface, developed using the TauDEM ArcGIS toolset (v5.3.7; [57]). These data along with average seasonal estimates of maximum air temperature and soil moisture were then computed for the Australian Alps. In small number of cases, individual pixels of the maximum air temperature and soil moisture rasters exceeded the maximum/minimum points on which the linear regression model had been developed, in which case they were forced to the maximum/minimum value. We then made seasonal estimates of CH_4_ fluxes for each of the ecosystems measured across the Australian Alps based on our linear regression model and data for soil moisture and temperature.

## 3. Results

### 3.1. Soil Greenhouse Gas Fluxes and Production at Depth

Air temperatures ranged from 3.1 to 28.3 °C, while the average soil temperature at 0–7 cm depth was 3.6 to 16.4 °C (Appendix A). Gravimetric water content on 17 February (when samples were collected for microbial community analyses) ranged from 0.25 to 0.79 for forest soils, 0.25–0.38 for grassland soils, and 0.9–4.66 g H_2_O g dry soil^−1^ for bog soils (Appendix A). Relative differences in soil temperature and moisture amongst soil/vegetation types during February were maintained across the three other sampling dates (Appendix A).

Mean soil-to-atmosphere CO_2_ fluxes (measured in situ) across the soils ranged from 2.5 to 17.4 mg CO_2_ m^−2^ h^−1^, in spring and summer, respectively (Figure 2A–D). Belowground, CO_2_ production (measured in the laboratory) decreased with depth. CO_2_ production thus dominated by emissions from the top 0–5 cm of soil that ranged from 72 to 2357 µg CO_2_ g^−1^ d^−1^. Contrastingly, at 25–30 cm depth, CO_2_ production ranged from 4 to 197 µg CO_2_ g^−1^ d^−1^. Forest and bog soils showed significantly greater CO_2_ production in summer than the grassland soil (*p* < 0.01, Figure 2E), ranging from 47 to 398% greater in forest soil or 60 to 282% greater in bog soil. There were less pronounced differences amongst soil types in autumn, winter, and spring.

In situ CH_4_ fluxes ranged from −413 to +778 µg CH_4_ m^−2^ h^−1^ (Figure 3A–D). Absolute fluxes, either positive or negative, were greatest in summer compared to other seasons. Patterns with depth were similar across three seasons. Forest and grassland soils mostly consumed CH_4_ throughout the year (Figure 3A–D), while bog soils were net producers of CH_4_ in three of four seasons (Figure 3A–C). Most soils (including bog) showed net CH_4_ consumption (across all depths; Figure 3E–H). In summer, we found large CH_4_ production in bog soils (3838 µg CH_4_ g^−1^ d^−1^) at 5–10 cm depth (*p* = 0.003), and moderate CH_4_ production at 25–30 depth ranging from 87 to 704 µg CH_4_ g^−1^ d^−1^ (Figure 3E). Expressed in CO_2_ equivalents, negative CH_4_ fluxes (CH_4_ consumption) offset CO_2_ emissions by 54 to 56% for grassland and forest soils, respectively (Table 1). CH_4_ emissions from bog soils, however, added an additional 40% of CO_2_ equivalents.

### 3.2. Archaeal and Bacterial Gene Abundance (qPCR) during Summer (17 February 2015)

There were significant differences in archaeal 16S rRNA gene abundances between soil types, with the bog soil showing one to two orders of magnitude greater abundances than the other two soils at multiple depths (*p* < 0.060, Figure 4A). Abundances of bacterial 16S rRNA genes were highly variable. Abundances decrease with depth from 2.3 × 10^11^ to 1.6 × 10^10^ copies per g dry soil (Figure 4B). There were no significant differences in abundance of bacterial 16S rRNA genes among soil types. The *pmoA* qPCR assay, which targets methanotrophs belonging to the Methylococcaceae and Methylocystaceae families, showed greatest abundances in bog soils (Figure 4C). The assay was not designed to detect the *pmoA* genes of USCα or *Candidatus* Methylomirabilis.

Across the soil/vegetation gradient, both CH_4_ and CO_2_ production at all depths was positively correlated with the soil-to-atmosphere flux of both gases (Appendix A). There was also evidence that this GHG production correlated with specific gene abundances across all soil types and depths. For example, the abundance of bacterial 16S rRNA genes was linearly and positively related to CO_2_ production (Appendix A); while the abundance of Euryarchaeota 16S rRNA was non-linearly and positively related to CH_4_ production (Appendix A). We did not find significant relationships between abundances of archaea and CO_2_, nor between bacterial abundances and CH_4_.

### 3.3. Microbial Community Composition and Diversity during Summer (17 February 2015)

Across all soils, Illumina sequencing resulted in 440,504 archaeal, 24,293,004 bacterial, and 2,258,405 *pmoA* sequences. Clustering soil communities across vegetation and depth showed distinct differences in bacterial 16S rRNA genes between the forest/grassland and bog soils, with forest and grassland soils being more similar to each other (Figure 5). In the bog soils, Betaproteobacteria (OTU-1850) and OTU-1522 were more abundant than in forest and grassland soils (Figure 5). Contrastingly, Rhizobiales (OTU-78) and OTU-71 differentiated the forest/grassland from bog soils. A few OTUs were uniquely abundant in grassland soils, like the Chloroflexi group (especially OTU-252, OTU-1043, OTU-249). Across all depths, bog soils had greatest diversity of 16S rRNA compared to the forest and grassland soils (Appendix A). More specifically, bog soils had 10% and 42% greater Shannon diversity and richness (H’ and S) than the forest and grassland soils (*p* < 0.001).

This study focused on methanotroph identification by using Illumina sequencing of either 16S rRNA or *pmoA* genes. Both 16S rRNA and *pmoA* gene sequence data revealed that the forest and grassland soils were dominated by USCα (Figure 6)—with nearly equivalent abundances between the two soil types. The highest USCα relative abundance approached 1% of all 16S rRNA genes, and decreased with depth. *Methylocystis* were the most abundant aerobic methanotrophs detected in bog soils, accounting for up to 0.3% relative abundance of 16S rRNA genes across all depths (Figure 6C). The same was true for *pmoA* genes with those of *Methylocystis* being the most abundant in the bog soils (Figure 6F). The 16S rRNA genes of Ca. Methylomirabilis, which are nitrite-dependent anaerobic methanotrophs, increased with depth in the bog to a maximum of 1.5% relative abundance (Figure 6C). The *pmoA* gene of Ca. Methylomirabilis is not detected by the *pmoA* assay, which is why it is notably absent from that analysis (Figure 6F). Illumina sequencing of the *pmoA* gene revealed distinct differences among all three soil types, but especially between forest/grassland and bog soils—USCα-dominated methanotroph populations in forest and grassland soils, whereas *Methylocystis*-dominated bog soils (Figure 6D–F). USCα were conspicuously abundant at the surface of just one bog site (Site B4, Figure 1). We noted that this site had a greater slope while the soils had 80–1200% greater sand/rock content than the other bog sites (Appendix A). We cautiously suggest that these features enhanced the habitat for USCα at this one bog site.

### 3.4. Patterns in Community Composition and Relationships with Soil Properties

Non-metric multidimensional scaling (NMDS), similar to the clustering observed in Figure 5, showed a distinct difference in bacterial 16S rRNA genes among soil types and relationship to soil properties (*p* < 0.05, Figure 7). Bacterial communities in all soils showed clear patterns with depth, but patterns with depth were most distinct in bog soils. Soil properties associated with organic matter (e.g., nitrogen, carbon, phosphorus, root biomass, DOC) all correlated well to surface (0–5 cm) bacterial communities across all vegetation types. CH_4_ production also correlated with depth of bog soils (Figure 7). Grassland community composition positively related to clay and Fe content of the soils, and negatively with soil moisture—these soils were the driest of the three soil types (Figure 7; Appendix A). Bacterial communities were distinct in bog soils, and unsurprisingly correlated to water content, but also Ba content (and greater CH_4_ production).

Forest *pmoA* gene profiles were relatively tightly clustered except for one sample at 10–15 cm depth, whereas they varied more across depths in the grassland and bog soils. (Appendix A). CH_4_ production did not correlate with the *pmoA* gene profiles, indicating that methanotroph community composition did not predict the methane production potential of the soil. Community composition of Euryarchaeota showed differences among soil types, especially between grassland and bog soils, but no trend with depth (Appendix A). Both CO_2_ and CH_4_ were positively related to soil organic matter, whereas clay was negatively related to the bog Euryarchaeota community composition. In contrast, the community composition in the grassland soils positively associated with clay content (Appendix A).

## 4. Discussion & Conclusions

### 4.1. Soil Microbial Community Composition and Links to Greenhouse Gas Fluxes

Adjacent soils, including their soil-forming factors (especially organisms—vegetation, and landscape position) showed distinct microbial community composition and abundance with depth. Bog soils tended to have the greatest abundance of archaeal 16S rRNA and *pmoA* genes (Figure 4A,C). At multiple depths, bog soils also had greater diversity of bacterial and methanotroph communities (Appendix A). Both of these findings are likely related to large concentration of total organic C in bog soils compared to the other two soils. Averaged across all depths, bog soils had 41 and 128% greater soil organic C than forest and grassland soils, respectively (Appendix A). Available sources of energy (e.g., as approximated by soil C) are typically related to the size and diversity of microbial communities [58,59,60]. Soil pH also strongly regulates diversity and composition of bacterial communities [60]. These soils, however, showed little variation in soil pH (range of 5.4 to 5.7; Appendix A), indicating that pH was not a discriminating factor here.

Bog soils produced more CH_4_ at shallow depths (5–10 cm, Figure 3E). This is surprising considering that water content was greater with depth, i.e., they were likely to have lower redox potential and therefore to be methanogenic. Increasing abundance of *Candidatus* Methylomirabilis with depth may be one cause for this lower measured production in the deeper horizons (Figure 6). Ca. Methylomirabilis are nitrite-dependent and oxidize CH_4_ via intracellular production of O_2_ from the dismutation of NO [22,61], and therefore will have consumed some of the CH_4_ and thereby lowered the net production. Their increased abundance deeper in soil profiles is commensurate with increased availability of nitrite and reduced exogenous O_2_ [62]. In addition, *Sphagnum* rhizoids are abundant in our bog soils at depths 0 to 10 cm (Appendix A). Rhizodeposition by *Sphagnum* could thus supply organic C for either acetoclastic or CO_2_ reduction to CH_4_ (methanogenesis) [63]. Fast rates of CH_4_ release from bog soils is mostly expected and was evidenced here (Figure 3E), and supported by greater methanogen abundances (approximated by Euryarchaeota 16S rRNA, Appendix A). Previous work showed methanotroph:methanogen gene expression ratios to be negatively correlated with rates of CH_4_ emission in two peat bogs in the United Kingdom [31]. Therefore, coupled activities of CH_4_ oxidation and CH_4_ production within microsites help explain net fluxes of CH_4_ [1,64,65,66,67,68].

Abundances of methanotrophs declined with depth in forest and grassland soils (Figure 6), similar to the findings from other studies [1,69]. USCα methanotrophs are associated with high rates of atmospheric CH_4_ oxidation in well-drained habitats lacking substantial endogenous methanogenesis [69,70,71,72,73]. However, our bog sites showed strong CH_4_ consumption and relatively high USCα abundance at the 0–5 cm depth during the summer when soil-to-atmosphere CH_4_ fluxes were greatest (Figure 3A,E and Figure 6). Thus, USCα abundance, greater O_2_ availability, and reduced methanogenesis in the top 0–5 cm all could contribute to the high capacity for aerobic CH_4_ oxidation at the surface of bog soils. Although Ca. Methylomirabilis were highly abundant at depth in bog soils, their per-cell CH_4_ oxidation capacity is less than more aerobic methanotrophs [62]. O_2_ exposure during the subsectioning of the soil cores could also have inhibited methanogenesis during the laboratory incubations. Finally, although aerobic methanotrophs can survive in low O_2_ environments, in part by energy generation via fermentation reactions and by using alternative electron acceptors for respiration [74,75], rates of CH_4_ oxidation are generally less than in most well-oxygenated surface soils.

Bog soils showed a distinctly greater abundance of *Methylocystis*, Ca. Methylomirabilis, and unclassified proteobacterial methanotrophs (Figure 6). *Methylocystis* typically proliferate at CH_4_ concentrations >40 ppm [76,77]. A recent study illustrated the dependence of low-affinity methanotrophs on greater supply of CH_4_, and that this can additionally trigger high-affinity activity during drought [78]. Our grassland and forest soils were dominated by USCα (Figure 6). USCα, are classified as high-affinity, with apparent K_m_ values of 0.01–0.28, compared to that of 0.8–32 for low-affinity methanotrophs [79]. Several studies now suggest that forest or grassland soils with strong potential for CH_4_ oxidation also have greater abundances of *Methylocystis* or USCα methanotrophs [80,81]. There is mounting evidence, including this study, that absence/presence of specific methanotrophic bacteria or methanotrophic community composition may be just as important as physicochemical regulators of net CH_4_ fluxes from soils [81,82,83,84]. Regulation of CH_4_ fluxes by physicochemical processes such as substrate diffusion [85,86,87] or labile C supply [88,89] is important, and can be difficult to tease apart from changes in methanotroph community composition [83,86].

Our study and others [2,27] show that surface soils, even in consistently wet soil profiles (Figure 3), can be sinks for atmospheric CH_4_ and act as a ‘filter’ for CH_4_ produced at greater depths [78]. Some estimates suggest that as much as 80–97% of endogenously produced CH_4_ at depth is consumed before reaching the atmosphere [2,27,28,29]. Disturbance of these surface soils could easily result in larger net CH_4_ fluxes. Collectively, complex production/consumption dynamics, rapid net CH_4_ uptake rates (Figure 8), and heavy reliance on surface soils for CH_4_ consumption in bog soils make a clear case for further research.

### 4.2. The Australian Alps: Unique Soils with Disproportionate CH_4_ Sink Strength

Across the globe, aerated upland soils provide a net CH_4_ sink that ranges from 7 to 100 Tg y^−1^ [79]. This equates to 15% of the total global CH_4_ sink [80,92,93]. The Australian Alps are restricted to the southeastern corner of mainland Australia and account for just 0.16% of the area of the continent (Appendix A). While our study area is a small fraction of this total, there is good evidence that the alpine and sub-alpine regions constitute a sink that is disproportionately large relative to other ecosystems (Figure 8). Using a combination of measures to estimate areal contributions of the different vegetation types, and extrapolating across the region (Table 2), we show that while comprising <0.03% of global forested and grassland ecosystems, these Australian ecosystems could represent a far greater proportion of global soil CH_4_ sinks [94,95,96]. These soils are clearly strong sinks for CH_4_ (Figure 8). We acknowledge that scaling and comparing annual CH_4_ flux estimates is prone to difficulties owing to the lack of observations [96]. Monitoring CH_4_ fluxes at high spatial and temporal resolution in remote locations remains a major challenge.

Forest and grassland sinks for CH_4_ also represent major offsets (in negative CO_2_-equivalents; 53% and 56%, respectively) to the total GWP of CO_2_ emissions. All soils consumed CH_4_ at 0–5 cm depth. This is consistent with other studies showing fast uptake of CH_4_ in surface soils where O_2_ is more available [1,97,98]. Even some lake sediments show similar CH_4_ oxidation profiles with depth to those recorded here [3]. However, many studies suggest that oxidation rates are greater at 5–10 cm than in the surface 5 cm [1,19,99,100,101]. Much of the variation in soil depth of maximum CH_4_ consumption depends on texture, organic matter and moisture content, as well as availability of labile C and inorganic N [88,89,99].

The native ecosystems studied here, with arguably disproportionate importance as CH_4_ sinks (Table 1 and Table 2; Figure 8), are amongst those most likely to be affected by changes in climate. Declines in CH_4_ uptake have recently been reported in several long-term studies of forest soils [102]. More frequent and intense forest fires are just one global change threat to these Australian Alps soils [103,104], with potential positive feedbacks to climate change. Ecosystem services provided by soils are critical to the planet [105], and agroecosystem soils receive much attention due to their proximal benefits (i.e., crop production). Globally, strong rates of atmospheric CH_4_ oxidation in alpine and sub-alpine regions are less obvious and less easy to value but are critical buffers against anthropogenic climate change.

## Figures and Tables

**Figure 1 microorganisms-09-00606-f001:**
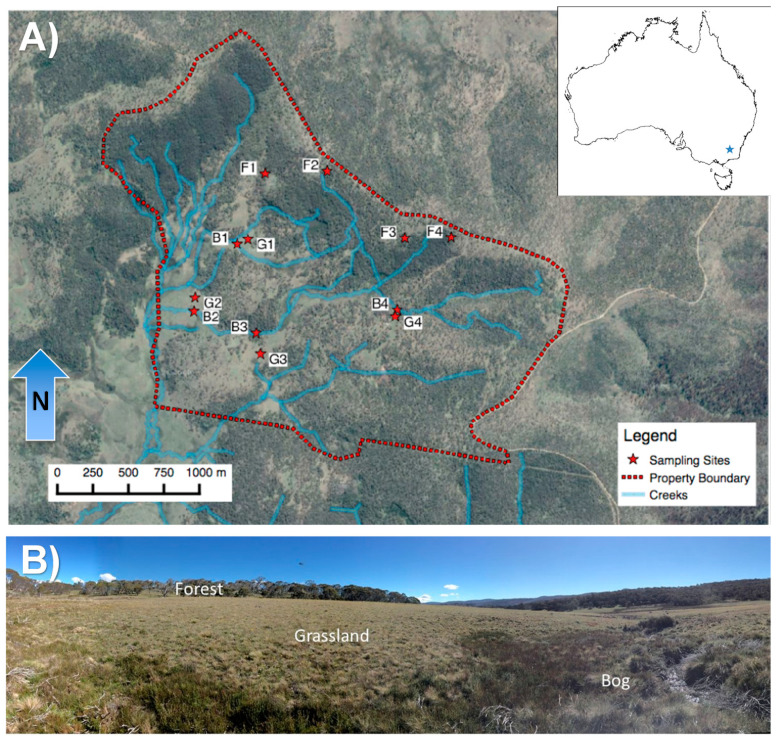
(**A**) Location of 548 ha experiment area (inset of Australia), and map of sampling sites within watershed and nearby streams. (**B**) Landscape-level photograph of the vegetation gradient from *Sphagnum*-dominated bog in foreground to eucalyptus-dominated forest in background. Abbreviations are F = forest, G = grassland, and B = bog. The numbers after the letter represent which transect the sampling location belongs to. Map and Image sources: L. Ingram andEsri, Maxar, GeoEye, Earthstar Geographics, CNES/Airbus DS, USDA, USGS, AeroGRID, IGN, and the GIS User Community. Photograph source: M.D. McDaniel.

**Figure 2 microorganisms-09-00606-f002:**
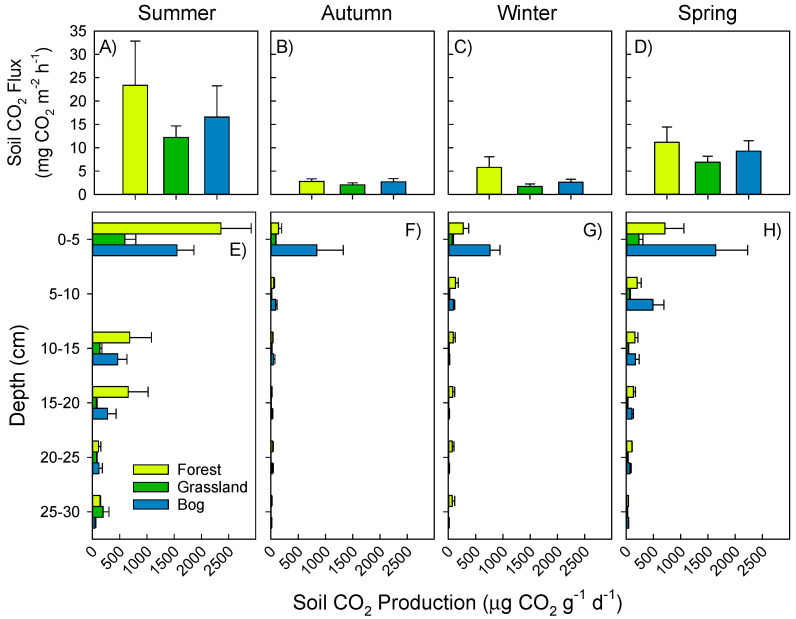
Soil CO_2_ fluxes (top panels, **A**–**D**) and production (bottom panels, **E**–**H**). Surface flux measurements and soils collected for production on 17 February (summer, **A**,**E**), 25 May (autumn, **B**,**F**), 22 September (winter, **C**,**G**), and 23 November (spring, **D**,**H**) in 2015. Mean and standard error shown (*n* = 4).

**Figure 3 microorganisms-09-00606-f003:**
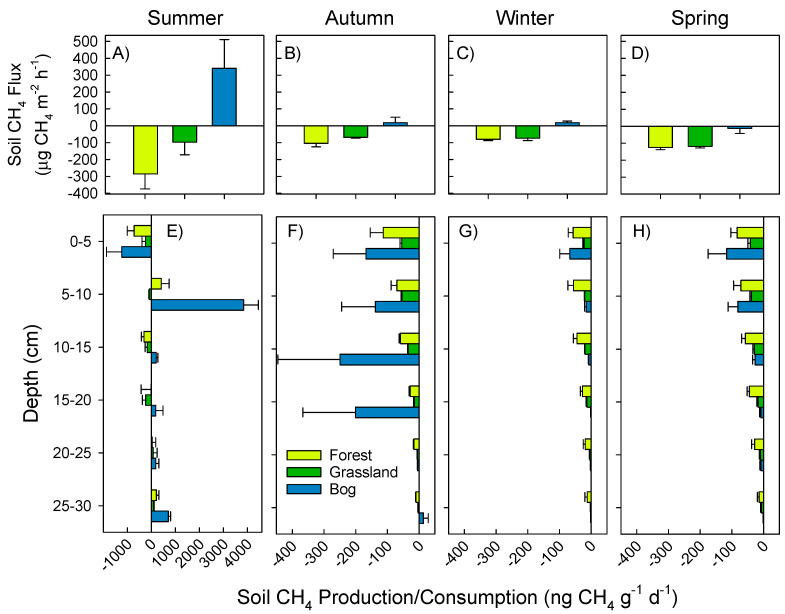
Soil CH_4_ fluxes (top panels, **A**–**D**) and production/consumption (bottom panels, **E**–**H**). Surface flux measurements and soils collected for production on 17 February (summer, **A**,**E**), 25 May (autumn, **B**,**F**), 22 September (winter, **C**,**G**), and 23 November (spring, **D**,**H**) in 2015. Mean and standard error shown (*n* = 4).

**Figure 4 microorganisms-09-00606-f004:**
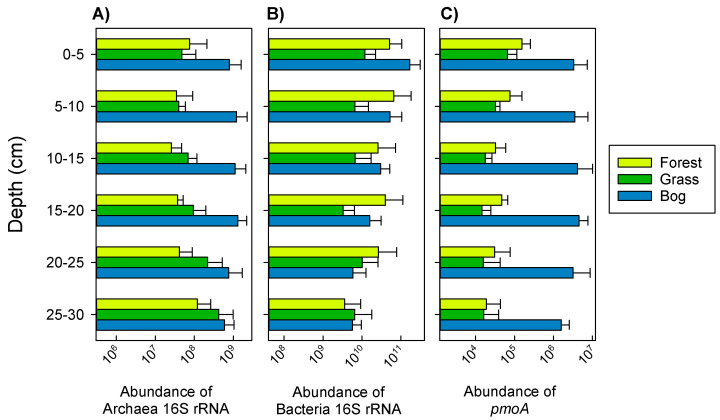
Abundance of archaeal (**A**) and bacterial (**B**) 16S rRNA genes, and *pmoA* (**C**) per g of dry soil. Means and standard error are shown (*n* = 4) for all samples.

**Figure 5 microorganisms-09-00606-f005:**
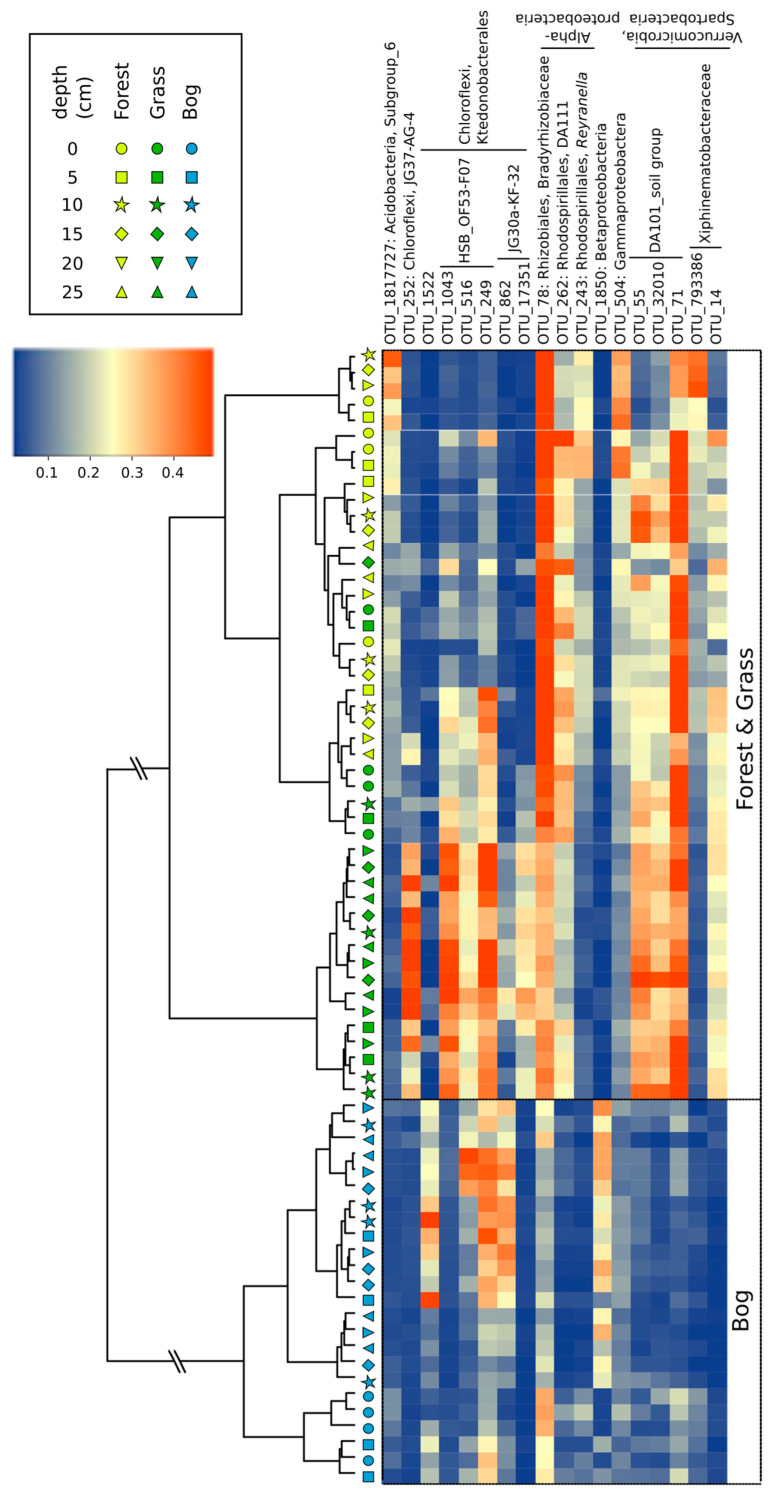
Heatmap of the OTUs derived from bacterial 16S rRNA genes. The OTUs with the highest loadings in a PCA analysis were selected. The samples and OTUs were clustered according to Euclidean distances between all Hellinger-transformed data. The taxonomy of OTUs was determined using the Silva classifier. The colored scale gives the percentage abundance of OTUs.

**Figure 6 microorganisms-09-00606-f006:**
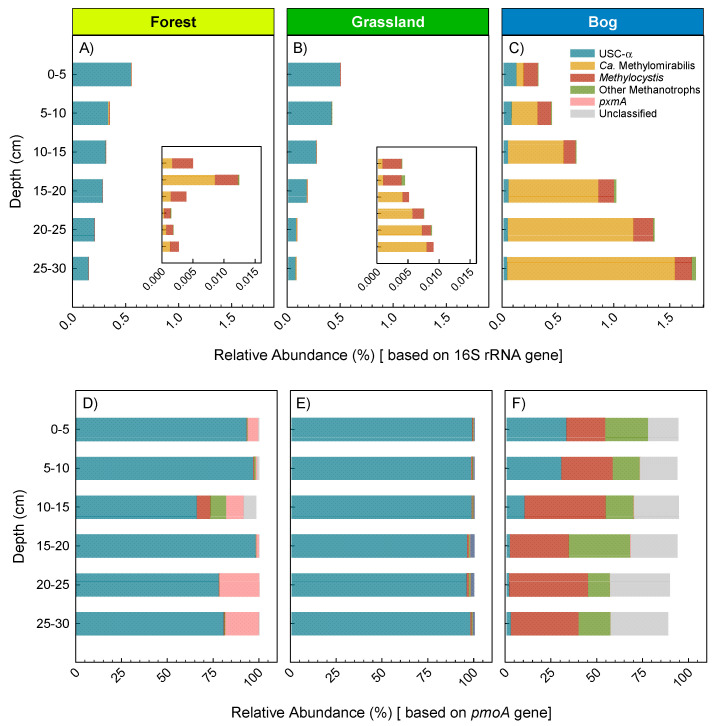
Dominant methanotroph groups detected in the forest (**A**,**D**), grassland (**B**,**E**), and bog (**C**,**F**) soils, based on relative abundance of 16S rRNA and *pmoA* genes, top and bottom panels, respectively. Inset graphs in A and B show abundance for each corresponding depth but at smaller *X*-axis scale. USCα was identified by blast as described in the methods. *Methylocystis* and Ca. Methylomirabilis were identified based on the Silva classifications. Other methanotrophs include *Methylomonas* and *Methylospira*. *pxmA* refer to *pmoA*-like genes of uncertain function found in the genomes of various methanotrophs.

**Figure 7 microorganisms-09-00606-f007:**
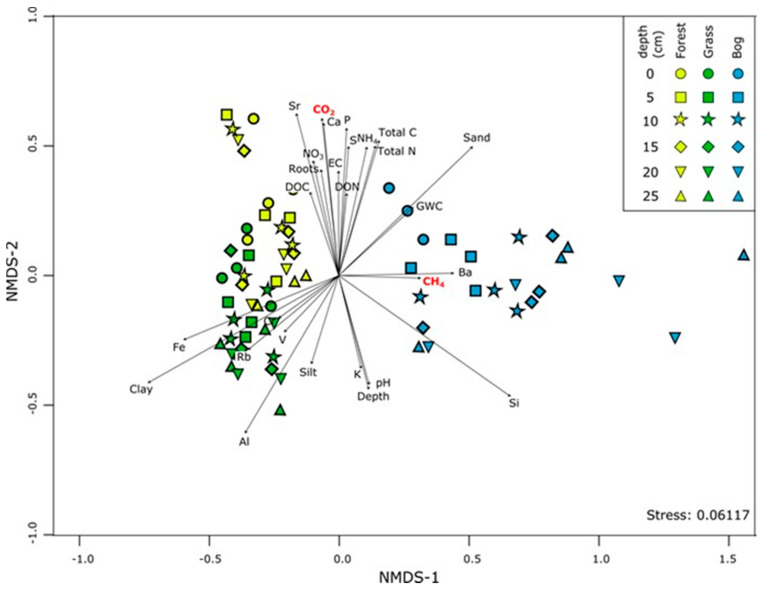
Nonmetric multidimensional scaling (NMDS) ordination of bacterial 16S rRNA communities based on the Bray–Curtis dissimilarity of community composition. Arrow vectors are environmental predictors (CO_2_, CH_4_, heavy elements, and other soil properties) that best fit onto the NMDS ordination space. Abbreviations: EC, electrical conductivity; DOC, dissolved organic carbon; DON, dissolved organic nitrogen; GWC, gravimetric water content; NO_3_, nitrate.

**Figure 8 microorganisms-09-00606-f008:**
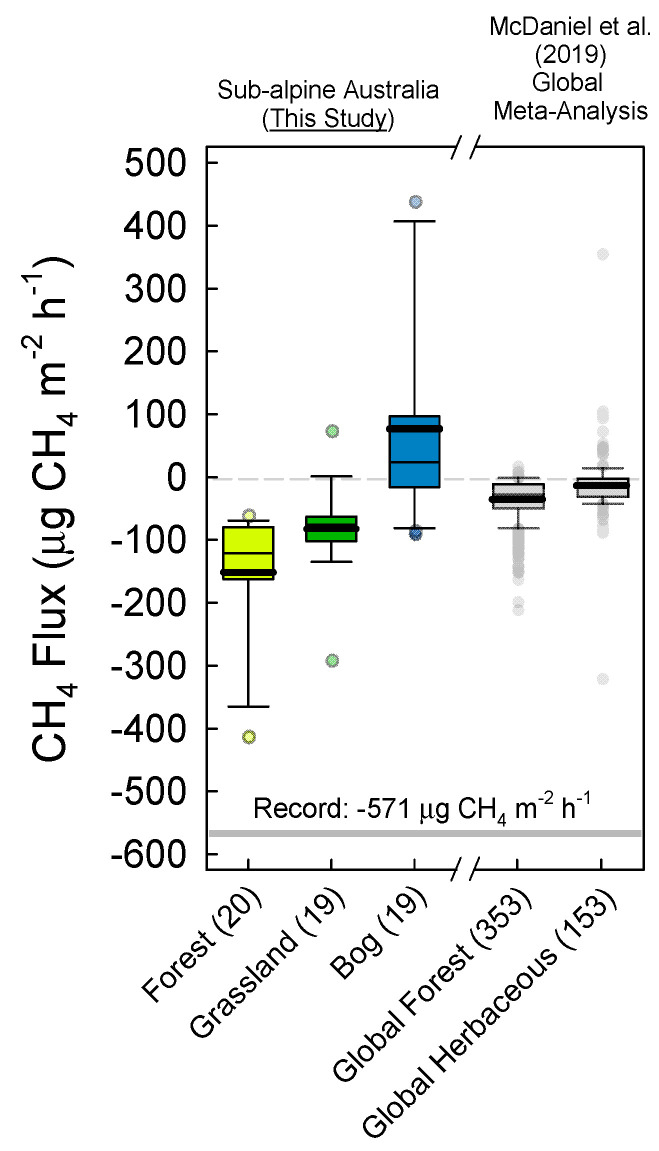
Hourly CH_4_ fluxes from this study’s forest, grassland and bog (from Figure 3A–D) soils compared to forest and herbaceous studies from a global meta-analysis [90]. The 10th and 90th percentiles are shown by bottom and top whiskers. The 25th and 75th percentiles are shown by the bottom and top of the box. Median is shown by the thin line, mean by the thick line, and outliers are circles. The number of measurements within each boxplot are shown in parentheses. Gray bar at −571 μg CH_4_ m^−2^ h^−1^ is the greatest CH_4_ oxidation rate (most negative flux) ever observed and published [91].

**Table 1 microorganisms-09-00606-t001:** Annual estimates (mean, standard error, and range) for net ecosystem flux of CO_2_ and CH_4_.

Ecosystem	% of 548 ha Watershed	Net Ecosystem Flux	
		CO_2_	CH_4_
		CO_2_-equivalents g m^−2^ y^−1^	
Forest	8–75	911 ± 165	−506 ± 22
Grassland	25–91	484 ± 50	−259 ± 38
Bog	1	646 ± 109	256 ± 82
		CO_2_-equivalents kg ha^−1^ y^−1^	
Total Watershed		5207 to 8031	−2762 to −4391

**Table 2 microorganisms-09-00606-t002:** Comparison of global and Australian Alps soil CH_4_ sink estimates.

Measurement	Values		
**Ecosystems Areal Coverage**			
Global forest + grassland ecosystems (M ha)		5100	
Areal coverage of Australian Alps—Appendix A (M ha)		1.23	
Fraction of Australian Alps to global forest + grassland (%)		0.024	
**Soil CH_4_ Sink Estimates**	**Low Estimate**	**Best Estimate**	**High Estimate**
Global CH_4_ soil sink ^†^ (Tg y^−1^)	−9	−30	−100
Mean annual Australian Alps CH_4_ sink ^‡^ (kg ha^−1^ y^−1^)	−4.2	−19.2	−33.2
**Australian Alps Contribution to Global CH_4_ Sink (%)**	**Low Estimate**	**Best Estimate**	**High Estimate**
Low estimate (−4.2 kg ha^−1^ y^−1^)	69	21	6
Best estimate, or our projection (−19.2 kg ha^−1^ y^−1^)	213	64	19
High estimate (−33.2 kg ha^−1^ y^−1^)	359	108	32

^†^ Low and best estimates from Kirschke et al. [94] and Saunois et al. [95]. Other estimates have a high estimate of −100 Tg y^−1^ [79]. ^‡^ Based on forest foliage imagery, soil temperatures and moisture estimates, and CH_4_ modeling described in Experimental Procedures. High and low estimate from +/− relative standard deviation from ecosystem means.

## Data Availability

Sequence data have been deposited in the NCBI Sequence Read Archive (SRA) under accession number PRJNA384296.

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
