# Peer review of "Disproportionate CH4 Sink Strength from an Endemic, Sub-Alpine Australian Soil Microbial Community"

_microorganisms, 2021, doi:10.3390/microorganisms9030606_

Round 1

Reviewer 1 Report

Major point

This is the type of a paper dealing with correlations of the GHG consumption/production with seasons, soil types and community structures. Overall, the study is solid and provides a set of data for a soil-vegetation gradient of a subalpine system in Australia. 

My main concern is, with so many conclusions driven about the communities participating in GHG production/consumption, data only on a single point are presented on the community structure. So we remain in the dark as to whether the communities dramatically change, or somewhat change with seasons, and whether such changes might be responsible for the changes in gas fluxes observed. I would like for the authors to present a good argument for why only one point was sampled for community structure.

Minor comments

L 48 You mean gammaproteobacteria here, not delta, correct?

L 55 I disagree with the statement 'evaded cultivation' they have not, you cite ref 25, which describes exactly that. 

Reviewer 2 Report

McDaniel and Hernandez et al. have studied the relationships between CO2 and CH4 production rates, consumption rates, overall fluxes and the microbial community members involved in methane cycling in forest soil, grassland soil and peat bog ecosystems. The study encompasses multiple sites within a region of the Australian “alps” and flux measurements cover several time-points to include all seasons.

I like the comparison of these ecosystem types and the focus on methanotrophs is justified. The methanogens falls on the side as they are only prominent in one of the ecosystem types. The coupling of methanotroph communities to production rates, fluxes and ecosystem types is interesting and the experimental setup fits well for this. However, the attempts to link production rates to fluxes and bacterial/archaeal numbers is not convincing, partly due to the results being unclear, and partly because the findings do not seem to have been critically assessed by the authors. The interpretation appear to be based exclusively on the p-value (please correct me if im wrong, but this is just the impression I get when I read the manuscript).

The overall community overviews are fine, as are the methanotroph overviews. I also like the overall CH4 flux comparison in the discussion but somehow think this should be moved to the results.

However, when finishing the results section, my immediate feeling was that there was too much emphasis on correlations and methanogens considering that there are no major findings related to this. This contradiction should be addressed, and can probably be fixed for an overall better paper. Especially the findings related to figure 5 should be toned down + be interpreted more critically.

My opinion is that here is a strong dataset on gas production, fluxes and methanotrophs, and that the sampling strategy and various gas measurements could support a very nice overview of gas exchange and methanotroph ecology. Perhaps investigating in more depth the correlation between abundances of various methanotroph groups at high resolution, gas fluxes and depths. This approach cold make use of OTU and phylogenetic cluster resolution to identify which groups of USCalpha and the other methanotroph groups dominate which environment and possibly something about the consequences. I think for this, the dataset may be appropriate.

Overall, my feeling is that the manuscript is two thirds finished, and toning down the more dubious, inconclusive or very general parts in favour of a deeper investigation of the ecology of the methanotrophs would serve the manuscript well.

Minor comments:

L24-27: The ending of the abstract is very general. I think there is potential for a more solid conclusion directly based on the results.

L35: Uptake of..?

L37: Heterotrophic decomposition or heterotrophy?

L38: Reference for the statement that autotrophy is slow?

L41-42: Maybe change sentence - the distance itself is not producing the difference

L55: Add “until recently”?

L62: I would add formate as it is often detected

L67: a bit odd logic here, little production is already evidence of little production. maybe rephrase sentence

L71-72: maybe hint towards why it is profoundly limited or more specifically what is limited

L79-81: Maybe explain why you think it is likely that they have a biogeochemical sensitivity to climate change and what that means

L129-132: There is something strange about this sentence, rewrite to be more clear

L138 – 140: It is not clear to me how you did the incubations to identify rates for the different layers.

L145: Here you need more info on type of GC, column, detector and method

L148-159: Im missing more details about the methods also here

L161-162: Rephrase, strange sentence

L166: Replace “was performed” with “was quantified”? Something is strange at least.

L296: methane or CO2 production? Or methane production in CO2 equivalents?

L296-300: Section has a bit sloppy language/annotation.

L299: Strange to say “exceptional” without reference point. Maybe statement better for discussion.

L317: Copies per g dry soil? Or is it wet weight? I cant see from the figure cause you didn’t put in the unit

L318-320: strange sentence - rephrase

Fig 4 legend: Why did you use standard error and not standard deviation?

L326:  at which depth?

L330: replace related with correlated?

Figure 5: My problem here is the sometimes low R squared and the fact that I don’t see the linear or other relationships although they are significant. It just seems as if there are to many variables influencing the variables to that it becomes impossible to draw useful conclusions using regression analysis. I suggest more care in concluding from these results, toning down their use in the paper and if you present as actual relationships and correlations you have to critically discuss so it is possible to follow your thoughts and consider the solidity.

Figure 6 legend: Why are the OTUs in the heatmap described as the most relevant? What does relevant imply?

L423-427: This sentence is a bit clumpy.

L430-431: Sentence is a bit weird. And also that nitrite is more available in deeper layers is a bit strange. Can be, but if you go deep enough is guess not anymore.

L435-437: Weird sentence “Bog soil showed..”

L441-445: This is getting very general - especially that methanogens are important in methane production.

L449-451: Also weird sentence – rephrase

L468-469: With this in mind, I suggest estimating the kinetics of the soil uptake using your data to compare between the sampling sites and ecosystem types. Can you also estimate methanotroph abundance combining relative abundanc of USC and qPCR to look into cellular oxidation rates? This would be very interesting.

L485-487: Very general

Fig. 8: Maybe rather include in results?

Reviewer 3 Report

McDaniel et al performed an interesting study connecting in situ gas fluxes, potential gas measurements and microbial (methanotrophic) abundance and community analyses at/in three different soil sites in Australia. Soil CH4 fluxes showed that forest and grassland soils were net CH4 sinks during the entire year (all seasons were measured once). In contrast, the bogs investigated were at least during summer CH4 sources on a net balance. The vertical soils profiles that were used to make incubation experiments confirmed the findings as the most indicated methane production was observed in bog soils (at a depth of 5-10 cm). The topic of the investigation is very interesting and the authors put much effort in analyzing the data. It is very much appreciated that the methanotrophic community was investigated in a separate sequencing run by amplifying the pmoA sequence only. Finally yet importantly, authors measured quantity of bacterial, archaeal and methanotrophic abundances in the vertical soil profile as well. The final part of the manuscript (discussions) was used to bring the story into a broader picture and relate the data with published numbers to estimate the Australian contribution to the global methane sink.

As mentionend, the story is of interest and the study well designed. Overall, the authors rather overtone the data and the reader is lost where the authors wanted to put the focus of the study (gas measurements, abundances, soil types, soil profiles, CO2, CH4, global relevance of Australian’s soils). This is even more obvious when additional data are presented in the discussion and the conclusion of the manuscript is about discussing the ecosystem service provided by Australian sites globally. The discussion somehow feels like a second manuscript. Still, it is important to emphasize that the combination of both gas measurements and microbiology is adding new value to the scientific knowledge in this field.

Major concerns

  1. Focus and structure of the manuscript: there is a huge dataset in the manuscript and authors tried to put into as much as they could which makes it not easy for the reader to get the story the authors want to tell. The authors are encouraged to focus the story and discuss the results more intensively (see comment above). Here is a suggestion how this could easily be arranged: Figure 5 (correlations) do not neither add much to the story nor are most of the correlations presented good correlations, e.g. Panel D is distorted by the 3 points of the bogs, Panel C is not surprising and not worth using figure space. More or less the same applies for Panel B. Figure 6, especially Panel B, is hardly discussed. Either delete it or spend some time and words in discussing this and also discuss why the most soil properties do not explain the prokaryotic community composition, so which parameters are missing that could have been relevant? High percentages of variance in the microbial communities of/in soils are often hard to explain. It is also not clear how much Figure 6 is contributing to the story line of the CH4 sink capacity of the soils. RE-think what you want to say and what is best to show then. Of course it is interesting how the entire prokaryotic community is structured and shaped by soil environmental properties, but in your big picture only when relating this information to the methanotrophic community? Do key-players in the prokaryotic community have partners in the methanotrophic community? Are there some associations?

Also think of exchanging Figure 7 with Figure S2 or combining them in a readable manner.

  1. Statistics: pls add significance values, e.g. Figure 2 (pls insert at least in the Panel A-D the significant differences of the in situ fluxes, same applies for Figure 3). Additionally, pls insert a panel showing the significant influence of depth in Figure 4 per site and microbial group or insert letters showing the significances.

  1. Precision and accuracy: authors are sometimes imprecise when discussing or presenting the results. Abundance and diversity is sometimes confused, 16S and bacterial (better say prokaryotic community and bacterial and archaeal abundances) and gives the reader the impression that the authors do not really know what they want to say; pls try to be accurate and precise because the data are good and are worth interpreting them thoroughly. Pls also frame the objectives of the study at the end of the introduction more precisely and balanced in terms of what is shown and discussed.

This also refers to the title of the manuscript. The title is smart and not fully incorrect, but it is not clear what is expected. A clearer title is appreciated. Also think of deleting the term “endemic” because it is hardly used (2 times) in the manuscript and is not increasing the meaning. Refers also to disproportionate

Minor concerns:

Line 23: it is not clear which soil site you are talking about when saying that Methylocystis sp were detcted at relatively low abundance.

Line 39: remove nitrous oxide

Line 40: remove N2O as this is not really part of your story

Line 79: give reference

Line 90: pls explain m asl for readership

Line 94: insert

Figure 1: can you improve the quality of the pictures?

Line 122: for how long (how many hours?)

Line 148: define which soils (from the collar? From the profile? Directly from the field?)

Line 149: add soil (soil sub-sample or sub-sample from soil)

Line 158: where are the results of heavy elements shown?

Line 176: literature is missing

Line 184: pls add positive controls. Did authors use negative and non-template controls? What was the positive control for the detection of the coverage of the pmoA genes?

Line 185: add amplicon

Line 187: exchange nucleotide with bp

Line 238: ggplot

Line 290: CH4

Line 296: 3,838 (or decide throughout the entire manuscript if to use thousand separators)

Line 323: pls give precise figure legend (this is to crude) and also pls indicate that the abundances are per gram dry weight (correct?); pls give standard deviation instead of standard error

Line 349: could OTU1850 not be classified on genus level?

Line 349: decide if to use “-“ between OTU and OTU number

Line 352: pls be precise à what do you mean with “across multiple depths”

Line 358: pls be precise and indicate from which test the most relevant OTUs were gathered (here as well and not in M&M only)

Line 362: delete stress and values in brackets (is already shown in the figure)

Line 380: pls rephrase “greater counts”

Line 389: pls explain inserts

Line 403: of course methane production and consumption are connected and the one causes the other pathway but be precise in these lines

Line 405: add CH4

Line 407-410: rephrase those sentences

Line 417: do you mean abundances?

Line 420 – Line 422: very much simplified explanation

Line 427: Methylomirabliis

Line 444: figure references seem to be mixed? Figure 5E does not exist, right? Fig 4C is not the right reference, is it?

Line 463: add literature

Line 467: order of listing the MOBs is not correct, right? UScalpha is much more abundant than Methylocytis? Methylocystis is very very little abundant, right which would also not really fit to the soil ecosystem studied….

Line 477: be precise and name physicochemical conditions from the literature

Line 488: CH4

Figures:

Figure 1 and 6: pls try to increase the quality of Figure 1A, 1B, 6A and 6B

Figure 6B: delete 16S rDNA headline

Figure S1: delete headline and give significances

Figure S2: pls explain pxmA

Round 2

Reviewer 2 Report

Manuscript has been appropriately revised and overall it looks good.

Reviewer 3 Report

The authors have clarified most of the points raised and have intensively worked on the revised manuscript. Unforatunately, the new Figure 6 is not readable which is probably an issue that occured while converting the document to a pdf file and is thus for sure easy to fix (colours and patterns are unfortunately not distinguishable at all).